# DNA Methylation Changes in Human Papillomavirus-Driven Head and Neck Cancers

**DOI:** 10.3390/cells9061359

**Published:** 2020-05-31

**Authors:** Chameera Ekanayake Weeramange, Kai Dun Tang, Sarju Vasani, Julian Langton-Lockton, Liz Kenny, Chamindie Punyadeera

**Affiliations:** 1Saliva & Liquid Biopsy Translational Research Team, Institute of Health and Biomedical Innovation, School of Biomedical Sciences, Queensland University of Technology (QUT), Queensland 4059, Australia; s.weeramange@qut.edu.au (C.E.W.); kai.tang@qut.edu.au (K.D.T.); 2Translational Research Institute, Queensland University of Technology (QUT), Queensland 4102, Australia; 3Department of Medical Laboratory Sciences, Faculty of Health Sciences, The Open University of Sri Lanka, Nugegoda 10250, Sri Lanka; 4Department of Otolaryngology, Royal Brisbane and Women’s Hospital, Queensland 4029, Australia; Sarju.Vasani@health.qld.gov.au; 5School of Medicine, The University of Queensland, Queensland 4006, Australia; lizkenny@bigpond.net.au; 6Metro-North Sexual Health and HIV Service, Queensland 4000, Australia; Julian.LangtonLockton@health.qld.gov.au; 7Department of Cancer Care Services, Royal Brisbane and Women’s Hospital, Queensland 4029, Australia; 8Central Integrated Regional Cancer Service, Queensland Health, Queensland 4001, Australia

**Keywords:** head and neck cancer, human papillomavirus, DNA methylation

## Abstract

Disruption of DNA methylation patterns is one of the hallmarks of cancer. Similar to other cancer types, human papillomavirus (HPV)-driven head and neck cancer (HNC) also reveals alterations in its methylation profile. The intrinsic ability of HPV oncoproteins E6 and E7 to interfere with DNA methyltransferase activity contributes to these methylation changes. There are many genes that have been reported to be differentially methylated in HPV-driven HNC. Some of these genes are involved in major cellular pathways, indicating that DNA methylation, at least in certain instances, may contribute to the development and progression of HPV-driven HNC. Furthermore, the HPV genome itself becomes a target of the cellular DNA methylation machinery. Some of these methylation changes appearing in the viral long control region (LCR) may contribute to uncontrolled oncoprotein expression, leading to carcinogenesis. Consistent with these observations, demethylation therapy appears to have significant effects on HPV-driven HNC. This review article comprehensively summarizes DNA methylation changes and their diagnostic and therapeutic indications in HPV-driven HNC.

## 1. HPV-Driven Head and Neck Cancer

Head and neck cancers (HNCs) are a diverse group of cancers arising in the upper aerodigestive tract, which includes the oral cavity, pharynx, larynx, nasal cavity, and paranasal sinuses [1]. Conventionally, the global burden of HNC has been mainly attributed to tobacco exposure and excessive alcohol consumption [2]. However, during the last two decades, the epidemiological landscape of HNC has been changed considerably, especially in developed countries, due to the continuous increase in human papillomavirus (HPV)-driven HNC [3,4,5]. Despite the available prophylactic vaccination program, a significant reduction in HPV-driven HNC incidence is not expected until 2060, necessitating further investigations to improve disease prevention, diagnosis, and treatment [3].

The oropharynx can be considered as the hotspot of HPV-driven cancers in the head and neck region, as the vast majority of these cancers arise in the palatine tonsils and base of the tongue [6,7]. These cancers have clinically and biologically distinct features from their HPV-negative counterparts in terms of their gene expression profiles, mutation burden, epigenetic profiles, and treatment response [8,9,10,11]. Despite the aggressive nature of HPV-driven HNCs, which often metastasize into nearby lymph nodes, these cancers are associated with a favourable prognosis, as they respond well to treatment [1]. However, the underlying biology of these cancers associated with these distinct features is poorly understood, especially the association between tumor attributes and epigenetic alterations.

Disruption of normal methylation patterns is one of the hallmarks of cancer. These changes are often associated with the tumorigenic process and tumor characteristics, such as tumor progression and metastasis [12]. However, HPV-driven cancers are unique, as viral proteins also contribute to these methylation changes. Studies conducted to interrogate the methylation landscape of HPV-driven HNC have revealed many inherent features. This review aims to summarise and evaluate the current state of knowledge on DNA methylation changes associated with HPV-driven HNCs and to discuss the diagnostic and therapeutic implications of these methylation changes.

## 2. Human Papillomavirus

Papillomaviruses are non-enveloped, icosahedral viruses, which belong to the *Papillomaviridae* family [13]. Over 400 papillomaviruses have been identified to date, and among them there are more than 200 types of papillomaviruses that can infect humans [14,15]. Papillomaviruses are epitheliotropic viruses, as their life cycles evolved to be dependent on the differentiation process of the epithelial tissue. Most of the papillomaviruses cause chronic asymptomatic infection with minimum effect on the host. However, several types (around 20) have evolved sophisticated mechanisms to promote cellular transformation [16]. These high-risk types are well known for their association with anogenital cancers and the increasing incidence of HNC. Among all HPV types, type 16 is associated with the majority of HPV-driven cancers, irrespective of the site [5].

The oncogenic potential of HPV is mainly attributed to HPV oncoproteins E5, E6, and E7 [17]. Through promoting cell proliferation and masking of the cell from the host’s surveillance mechanisms, these proteins allow cells to accumulate genetic and epigenetic changes over time, leading to oncogenesis [18]. These are multipurpose proteins that can interact and manipulate many cellular pathways [18]. Among the interactions, E7 protein’s ability to bind with retinoblastoma protein in order to hinder sequestration of E2F transcription factors, leading to transcriptional activation; and E6 protein’s ability to downregulate p53 in order to avoid cell cycle arrest and apoptosis are considered to play central roles [19].

Moreover, HPV oncoproteins E6 and E7 are also known for their ability to interact with cellular epigenetic machinery [20]. These interactions further promote genetic and epigenetic instability, and facilitate oncogenesis and tumor progression. Studies conducted to investigate epigenetic changes in HNC have detected many changes appearing in the epigenome. Among them, DNA methylation is one of the most studied epigenetic markers due to its possible utility in disease identification and treatment interventions. Early diagnosis of HNC has been one of the major concerns of the studies performed to date. Furthermore, DNA methylation has been identified to be involved in tumorigenesis, tumor progression, and metastasis across many types of cancer, rendering it a potential avenue for treatment interventions [21].

## 3. DNA Methylation

The covalent modification of bases in the DNA strand by the addition of methyl groups is referred to as DNA methylation. In eukaryotes, DNA methylation primarily involves cytosine [22,23]. In higher-order animals such as vertebrates, DNA methylation, when present, almost exclusively presents in the 5’ position of the cytosine ring [10,22,24]. Furthermore, cytosine methylation predominantly appears in cytosine bases, which are positioned 5’ to guanines (CpG dinucleotides) in the DNA strand [25,26].

Cytosine methylation is known to play a crucial role in genome stabilization. It is involved in gene imprinting, X chromosome inactivation, and repression of transposons and retroviral sequences in the genome [25]. In synchronization with histone modification, cytosine methylation orchestrates gene expression [26]. In most of the genes, regulatory sequences, such as promoters and enhancers, overlap with CpG islands [23,27]. These are the genes that are vulnerable to methylation-associated gene regulation [23]. Under normal conditions, most of the active and inactive genes have unmethylated regulatory sequences, indicating that DNA methylation is not the sole system of annotating genetic information [25]. However, it is consistently observed that cytosine methylation involving CpG islands in the regulatory sequences leads to transcriptional silencing of genes, and active genes often have hypomethylated or unmethylated regulatory sequences [23,28].

DNA methyltransferases (DNMT) are the enzymes that catalyze DNA methylation and are involved in both the establishment and maintenance of DNA methylation. DNMT1 is responsible for the transfer of methylation patterns to the newly synthesized DNA strand following DNA replication [29]. DNMT3A and DNMT3B, on the other hand, are involved in de novo methylation [30,31]. However, it has been shown that DNMT3A and DNMT3B are also required for methylation establishment and maintenance in addition to DNAMT1 [32]. In contrast, ten eleven translocation (TET) proteins counter the activity of DNMTs. TET1, TET2, and TET3 are capable of demethylating cytosines through a series of reactions initiating from oxidizing 5-methylcytosine to 5-hydroxymethylcytosine (5hmC) [33,34]. Moreover, studies indicate that activation-induced cytidine deaminase (AICDA) and thymine DNA glycosylase (TDG) are also involved in the demethylation process [35,36].

At the functional level, interpretation of the established epigenetic annotation involves many protein factors, such as the methyl-CpG-binding domain (MBD) family of proteins [26,37]. These factors, when bound to methylated DNA, are capable of restricting the access for transcription factors causing downregulation of the gene expression [38]. Furthermore, members of MBD family proteins are capable of interacting with histone modification enzymes, promoting a more closed chromatin structure in order to restrict the access for the transcription machinery [39,40].

## 4. DNA Methylation and Cancer

Disruption of DNA methylation patterns is often observed in cancer. In many cancer types, it has been observed that the genomes of cancer cells undergo an overall decrease in the level of 5-methylcytosines, while having increased 5-methylcytosines in the regulatory regions [41,42]. In certain instances, these changes, resulting in the inactivation of tumor suppressor genes and activation of oncogenes, act as alternates or complementary mechanisms to gene alterations, promoting carcinogenesis [43,44]. Moreover, certain methylation changes may enhance tumor progression and aggressiveness by promoting cell proliferation, disruption of cellular communication, promoting cell migration, downregulation of apoptosis, etc. [43,45].

## 5. DNA Methylation in HPV-Driven HNC

Similar to other cancer types, HPV-driven HNCs also reveal alterations in their methylation profiles [20,46]. However, due to the fact that HPV oncoproteins E6 and E7 are capable of interfering with cellular DNA methylation machinery (Figure 1), DNA methylation becomes more intricate, thus complicating interpretation in HPV-driven cancers [20].

It has been shown that global cytosine methylation levels are altered when p53 is inactivated [47,48]. Wild-type p53 is capable of binding and repressing the DNMT1 promoter in cooperation with specificity protein 1 (Sp1) [47]. As a result, when p53 is deactivated, DNMT1 levels tend to increase [47]. Suppression of p53 by HPV E6 is one of the main effects of HPV activity. In addition to all the other detrimental outcomes of p53 inactivation, this also results in DNMT1 upregulation [49].

The ability of HPV E7 protein to promote E2F activity by downregulating retinoblastoma protein (pRb) also promotes DNMT1 transcription, as E2F transcription factors have the ability to activate DNMT1 promoter [50,51]. Furthermore, HPV E7 also has the ability to bind with DNMT1 using the CR3 zinc finger domain located in the C terminal and to enhance the activity of DNMT1 directly [20].

## 6. Alterations in Methylation Patterns in HPV-Driven HNC

Studies conducted to investigate methylation patterns of HPV-driven HNC have demonstrated the ability of HPV oncoproteins to promote DNMT activity. Many of these studies have also revealed a significant increase in methylation levels, especially in the promoter regions, in HPV-driven HNC [46,52]. Moreover, transfection studies have confirmed the ability of HPV oncoproteins to promote DNA methylation [53]. However, certain genes have also been reported to be hypomethylated in HPV-driven HNC compared to non-HPV-driven HNC (Table 1).

## 7. Commonly Reported Differentially Methylated Genes in HPV-Driven HNC

### 7.1. Genes Involved in Cell Cycle Regulation and Programmed Cell Death

#### 7.1.1. *CDKN2A*

Changes in methylation patterns of the *CDKN2A* gene have been reported in several studies. This gene, positioned at 9p21.3, encodes two regulatory proteins, namely p16INK4a and p14ARF [76]. Both these proteins function as cell cycle regulators and are considered as important tumor suppressors; p14 ARF is involved in stabilization of p53 by downregulating MDM2 mediated p53 ubiquitination [77], while p16INK4a is a cyclin-dependent kinase (CDK) inhibitor involved in the retinoblastoma (pRb) pathway [78]. Genetic and epigenetic alterations in the *CDKN2A* gene resulting in expression changes have been observed in many types of cancer [79,80,81,82]. However, *CDKN2A* expression is considered even more imperative in HPV-driven HNC, as p16INK4a expression itself is considered as an independent prognosticator of HPV-driven oropharyngeal cancer (OPC) [83]. HPV-driven cancers usually present with higher p16INK4a levels attributable to E7-mediated repression of the pRB pathway [19,84]. Thus, in HPV-driven OPC, the *CDKN2A* gene is expected to be active and to have hypomethylated or unmethylated regulatory sequences.

Several studies have reported lower methylation levels in the *CDKN2A* locus in HPV-driven HNC compared to non-HPV-driven HNC [11,65,70]. This finding is not only seen in tumor tissue; hypomethylation in *CKND2A* has also been reported in salivary samples of HPV-positive HNC patients compared to healthy controls [58].

In contrast to these findings, several other studies have suggested that the gene is hypermethylated in HPV-driven HNCs compared to their non-HPV-driven counterparts. Schlecht et al. (2015) reported hypermethylation in non-promoter CpG islands in the *CDKN2A* gene compared to adjacent normal mucosa [1]. Swangphon et al. (2017) and Choudhury et al. (2015) reported *CDKN2A* promoter hypermethylation in HPV-positive HNC compared to normal control tissue [56,60]. Similarly, cervical cancer studies and penile cancer studies have reported hypermethylation in the *CKDN2A* locus compared to precancer [85,86,87,88,89]. Some studies have also reported *CDKN2A* hypermethylation in HPV-driven HNC in subjects with decreased P16 expression [56,64]. A similar observation has also been reported in a penile cancer study [89]. Despite these findings, some other studies that have been conducted to investigate *CKDN2A* methylation in HPV-driven HNC have not detected any significant methylation changes, indicating the heterogeneity of *CKDN2A* methylation patterns in HPV-driven cancers [67,72].

#### 7.1.2. *RASSF1*

The Ras association domain family 1 (*RASSF1*) gene located in the 3p21.3 position encodes for several protein isoforms containing a Ras association domain [90]. Among these isoforms, *RASSF1A* is considered to have tumor suppressor properties and has been shown to be silenced by genetic and epigenetic mechanisms in several types of cancer [90,91]. The *RASSF1* gene has been reported to be hypermethylated in HPV-related cancers, such as cervical cancer and penile cancer [88,92,93]. Consistent with these observations, Choudhury et al. (2015) reported promoter hypermethylation in the *RASSF1* gene in HPV-driven HNCs compared to non-HPV-driven HNCs [60]. However, this contradicts the findings of Colacino et al. (2013) and Dong et al. (2003), who reported hypomethylation of this gene in HPV-driven HNCs compared to their non-HPV-driven counterparts [65,75].

#### 7.1.3. *CCNA1*

The *CCNA1* gene at 13q13.3 encodes for cyclin A1 protein. Even though this protein is not the major A-type cyclin involved in the cell cycle, it has been shown in murine models that its absence slows down the cell cycle in somatic cells [94,95]. Furthermore, together with CDK2, cyclin A1 has been shown to regulate double-strand break repair in DNA [96]. Promoter hypermethylation in *CCNA1* gene has been reported in HPV-driven HNCs compared to non-HPV-driven HNCs in several studies [11,65,67]. Furthermore, promoter hypermethylation of the gene has also been reported in several other cancers, such as cervical cancer and nasopharyngeal carcinoma [97,98,99,100].

### 7.2. Genes Involved in Cellular Adhesion and Communication

#### 7.2.1. Cadherin Family Genes

The cadherin family of proteins is considered as the major class of cellular adhesion molecules, which are known to play crucial roles in cell signalling and communication, especially in solid tissues [101]. Proper cellular adhesion and communication ensure cellular coordination, which is critical to normal cellular differentiation, tissue morphogenesis, and homeostasis. The weakening of cellular adhesion associated with genetic and epigenetic changes in cadherin genes is often observed in epithelial cancers [102]. Methylation changes in cadherin genes have been reported in several HNC studies. Promoter hypermethylation of the *CDH8* gene located at 16q21, encoding a type II cadherin, is reported in several studies [46,53,59]. Promoter methylation of the same gene has also been reported in cervical cancer [103]. Furthermore, promoter hypermethylation of several other cadherin genes, including *CDH 18, CDH15, CDH13, CDH19*, and *CDH23*, has been reported by Lechner et al. (2013) in HPV-driven HNC [53]. Hypermethylation in the *CDH11* locus also has been reported in HPV-driven HNC in a different study [65]. Similar methylation patterns in cadherin genes, such as *CDH1* and *CDH13*, have been reported in cervical cancer [104,105,106,107].

Moreover, promoter hypermethylation of several protocadherin genes has also been reported in the literature. Among them, the protocadherin beta 11 (*PCDHB11*) locus has been reported to be hypermethylated in HPV-driven HNC compared to non-HPV-driven HNC [53,59]. Furthermore, hypermethylation in several other protocadherin genes, namely *PCDH8, PCDH9, PCDHB3, PCDH10, PCDH15, PCDHB1, PCDHB4,* and *PCDHB15*, has also been reported in HPV-driven HNC (53). When other HPV-associated cancers are considered, promoter methylation in *PCDHA4, PCDH10,* and *PCDHA 13* has been reported in cervical cancer [108,109].

#### 7.2.2. *ITGA4*

Integrin alpha 4 (*ITGA4*) gene has also been reported to be hypermethylated in HPV-driven HNC in a few studies (46, 55). *ITGA4* gene has also been reported to be hypermethylated in cervical cancer [110]. Many other non-HPV-driven cancer studies have reported similar findings [111,112]. The gene positioned at 2q31.3 encodes for alpha 4 peptides, which function as crucial integrins in association with beta 1 or beta 7 peptides [113,114]. These integrins mediate cellular adhesion, migration, and signalling [115]. Alpha 4 integrins are involved in the activation and function of hemopoiesis and leucocyte trafficking, playing an essential role in immunity [116]. Furthermore, alpha 4 integrins have also shown to be involved in the immune response against viral infections [117]. If the hypermethylation reported in the methylation studies is gene silencing in nature, it may have contributed to diminished viral clearance, leading to prolonged HPV infection. However, further studies are necessary to elucidate the existing association.

### 7.3. Genes Involved in Cellular Migration and Tumor Progression

#### 7.3.1. *TIMP3*

Matrix metalloproteinases (MMPs), one of the families of natural endopeptidases, are capable of degrading extracellular matrix components and are known to play a significant role in tumor progression [118]. Tissue inhibitor of metalloproteinase (*TIMP*) acts as the natural inhibitor of MMPs and is known to play a role as a tumor suppressor [119,120,121]. Promoter methylation of the *TIMP3* gene located at 22q12.3 has been reported in a few studies in HPV-driven HNCs compared to non-HPV-driven HNCs [52,67].

#### 7.3.2. *ELMO1*

Engulfment and cell motility 1 protein (*ELMO1*), encoded by the *ELMO1* gene located at 7p14.2-p14.1, is known to promote cellular migration [122]. This has been linked to increased invasion and metastasis in several types of cancer [122,123]. However, methylation studies report the gene to be hypermethylated in HPV-driven HNC [55,59]. It has not been clearly reported whether these hypermethylated CpGs are located in the promoter region of the gene. However, promoter methylation in the gene has been reported in cervical cancer [124].

### 7.4. Genes and Non-Coding Regions with Unknown Association with Carcinogenesis 

#### 7.4.1. *MEI1*

Lechner et al. (2013) reported the promoter of the *MEI1* located at 22q13.2 to be hypomethylated in HPV-driven HNCs compared to non-HPV-driven HNCs. Even though exact sites of methylation have not been specified, hypomethylation in the same locus has been reported Worsham et al. in HPV-driven HNC [59]. The gene encodes for meiotic double-stranded break formation protein 1, which is known to involve in meiosis [125]. A functional association between the gene and cancer has not been reported as of yet.

#### 7.4.2. *LINE1*

Long interspersed nuclear element 1 (*LINE1*) is one of the most abundant types of retrotransposons accumulated in the human genome, which accounts for approximately 17% of the total DNA [126]. Most copies of *LINE1* have been inactivated by truncations, rearrangements, and mutations, and only several copies retain their ability for retrotransposition [126,127]. DNA methylation plays a key role in silencing these competent *LINE1* sequences [128]. Hypermethylation of *LINE1* region, in comparison to HPV-negative HNC, has been reported in HPV-driven HNC [11,68,69]. However, *LINE1* methylation in HPV-driven HNC compared to healthy controls is not clear, as it has been reported that HPV-negative tumors tend to have hypomethylated *LINE1* regions [69].

## 8. HPV DNA Methylation in HPV-Driven HNC

The involvement of DNA methylation in gene imprinting, X chromosome inactivation, and repression of transposons and retroviral sequences indicates that DNA methylation fundamentally intends to silence the undesirable or harmful sections of the genome in order to achieve genomic stabilization [25]. Possibly for the same purpose, HPV genomes within the host cells become targets of the cellular methylation machinery.

The double-stranded DNA genome of HPV comprises approximately 8000 base pairs [13]. The genome can be divided into 3 regions, namely the early region (E), late region (L), and upstream regulatory region (URR) or long control region (LCR) (Figure 2) [15,129]. The HPV genome has only two common promoters, an early promoter and a later promoter [130]. The early promoter, which is located in the viral LCR, is responsible for the transcription regulation of HPV early proteins, including major oncoproteins E6 and E7 [130,131].

Similar to promoter methylation in the host genes, DNA methylation in the viral LCR may downregulate the transcription of early viral genes [132]. This downregulation is often associated with the deregulation of transcription factor binding [132,133,134]. Binding of methyl CpG binding proteins to the methylated sites is also known to obstruct transcription factor binding [132]. However, LCR methylation does not necessarily cause transcriptional silencing. Instead, selective methylation can even cause overexpression of early viral proteins [135].

E2, the major regulatory protein of HPV, is capable of controlling the transcription of HPV early proteins through a mechanism involving negative feedback control [136,137]. In most mucosal HPVs, LCRs contain 4 E2 binding sites (Figure 3) [138]. Among those binding sites, E2BS1 located towards the 5’ end of the LCR has the highest affinity for the E2 protein. E2BS1 recruits E2, even at lower concentrations, and this binding activates the viral early promoter, enhancing the transcription of HPV early proteins, including E6, E7, and E2 itself [135,138,139]. As a result, E2 levels increase and E2 starts to bind the other E2 binding sites towards the 3’ end of the viral LCR, which have comparatively lower affinities to E2 [135,138]. This binding, especially with E2BS3 and E2BS4, represses the early promoter, hindering the transcription of early proteins [135]. E2BS3 and E2BS4 are located in close proximity to SP1 and transcription factor II D (TFIID) binding sites, and binding of E2 to these sites displaces SP1 or TFIID, repressing the transcription [134].

Alterations in E2 binding sites or E2 protein itself may disrupt this regulation mechanism. For instance, changes in E2BS3 or E2BS4 disrupting the E2 binding to these sites would result in overexpression of HPV early proteins. HPV E2 binding sites have a common palindromic sequence of ACCG(N)_4_CGGT [135]. E2 binds to these DNA motifs in the form of a dimer, with the DNA binding domain positioned at the carboxy-terminal of the protein [144]. However, CpG methylation in the binding sites hinder E2 binding to the site, disturbing E2 mediated viral early promoter regulation [145]. As such, methylation in E2BS3 or E2BS4 may lead to disruption of E2-mediated transcriptional regulation, leading to overexpression of viral oncoproteins, increasing the odds of tumorigenesis [131,146]. Studies conducted to investigate LCR CpG methylation in HNC have reported varying degrees of methylation levels (Table 2) [147,148,149]. These findings suggest that CpG methylation in LCR is not common to all HPV-driven HNC. Similar findings have also been reported in cervical cancer studies [150,151,152]. LCR methylation, especially CpG sites at E2BS3 and E2BS4, seems to be an alternative pathway of losing E2 control, leading to oncoprotein overexpression. Thus, it is not necessarily present in all the HPV-driven cancers.

Not only in viral LCR, CpG methylation has also been observed in other CpG sites of the HPV genome. In cervical cancer, viral *L1* and *L2* genes have been reported to have higher methylation levels in cancers compared to low-grade precancer [153,154]. HPV DNA methylation in these sites has not been comprehensively investigated in HPV-driven HNC. However, available literature suggests similar methylation patterns in HNC [54,155]. However, a functional relationship between CpG methylation of these sites and tumorigenesis has not been identified to date. It can be speculated that this type of methylation may result from long term HPV infection with ineffective replication of the late viral genes.

## 9. Diagnostic Implications

Even though studies have proposed methylation signatures characteristic to HPV-driven HNC, these signatures are not consistent among studies. Except for a few genes, most of the genes reported to be differentially methylated have been reported only in a single study. Thus, the methylation profiles of HPV-driven HNCs appear to be considerably heterogeneous. However, extrapolating the results from different studies and drawing conclusions are challenging issues, as methodologies of the studies are considerably dissimilar in terms of identification of the HPV status of the tumor, evaluation of the methylation status, and the comparisons used.

The majority of studies have reported on methylation status by comparing HPV-driven and non-HPV-driven HNCs, an approach that does not allow identification of methylation changes from normal methylation patterns. Furthermore, these comparisons may erroneously identify hypermethylation in one group as hypomethylation in the other group. Despite these challenges, several genes and loci have been identified to be differentially methylated by independent studies. However, the available information is not sufficient to make firm conclusions. As such, more comprehensive multicentered studies with uniform methodologies are necessary to identify diagnostic methylation targets.

The diagnostic value of HPV DNA methylation patterns also needs to be further investigated. Cervical cancer studies have suggested that methylation changes in HPV *L1* and *L2* genes can be used to discriminate cervical cancer from low-grade precancer [153,154]. Even though HNC studies have also reported similar findings, these associations have not been comprehensively investigated [54].

The majority of the studies have considered only HPV16, as it is the most common type involved in HPV-driven cancers, irrespective of the site. Studies that have considered other high risk (HR-HPV) types have not reported separate data for these HR-HPV types, possibly due to the lower number of samples. Even though these findings are generalized for HR-HPV, characteristic features may exist in cancers caused by different HPV types. However, such interpretations are not possible due to the paucity of available data.

## 10. Therapeutic Implications

Despite the differences between studies in terms of their methodologies many studies have shown that HPV-driven HNCs have higher overall methylation levels compared to their HPV-negative equivalents [11,46,52,53]. Methylation changes have been reported in genes that are known to be associated with tumorigenesis, tumor progression, and metastasis across many cancer types. Furthermore, methylation changes in the HPV genome leading to oncoprotein overexpression have also been identified. Presumably, these methylation changes may play a role in HPV-driven HNCs, at least in certain instances.

Consistent with these findings, demethylation therapy has shown to be effective against HPV-driven HNCs. In both HPV-positive HNCs and cervical cell lines, demethylating agent 5-aza-2′-deoxycytidine (DAC) has shown efficacy in downregulating HPV E6 and E7 expression [156]. In addition to hindering oncoprotein expression, demethylation treatment has been found to increase p53 and p21 protein levels and tumor suppressor microRNA 375 levels, resulting in an overall decrease in cancer cell growth and survival [156]. Similar observations have been reported in a different study using demethylation agent 5-azacytidine (5-aza) [157]. In vivo studies have confirmed in vitro findings, with similar responses for demethylation therapy by HPV-driven HNC. A study conducted using xenografted tumors in nude mice reported tumor growth inhibition [157]. The same study also reported a marked decrease in HPV oncoprotein expression, p53 stabilization, and upregulated caspase activity in HPV-driven HNC patients [157].

Furthermore, an increased interferon response in certain HPV-driven HNC cell lines and a decrease in matrix metalloproteinases (MMP) expression in both mouse models and HPV-driven HNC patients following demethylation therapy were reported in the same study [157]. There is a possibility that these changes are associated with the activation of host genes following demethylation therapy. For instance, it has been reported that promoter methylation of the *TIMP3* gene, which encodes a natural metalloproteinase, is commonly seen in HPV-driven HNC [52,67]. Activation of the TIMP3 gene following demethylation therapy would result in an increase in the TIMP3 level, eventually inhibiting MMP endopeptidases.

Several clinical trials have been conducted to evaluate the effectiveness of demethylation therapy on HNC [158]. A specific clinical trial to assess the effectiveness of demethylation therapy on HPV-driven HNC is currently underway (NCT02178072).

## 11. Conclusions

Many changes in the host’s methylation landscape have been observed in HPV-driven HNC. However, the proposed methylation signatures are not consistent among studies; except for a few genes and loci, the vast majority have not been repeatedly reported in studies. Despite these inconsistencies, many studies have reported overall higher methylation levels in HPV-driven HNC. Furthermore, several genes that are known to be associated with tumorigenesis, tumor progression, and metastasis have been reported to be differentially methylated in HPV-driven HNC. In addition to methylation changes in the host genome, methylation changes in the HPV genome have also been reported. Consistent with these findings, demethylation therapy has been shown to have significant effects on HPV-driven HNC cancer.

## Figures and Tables

**Figure 1 cells-09-01359-f001:**
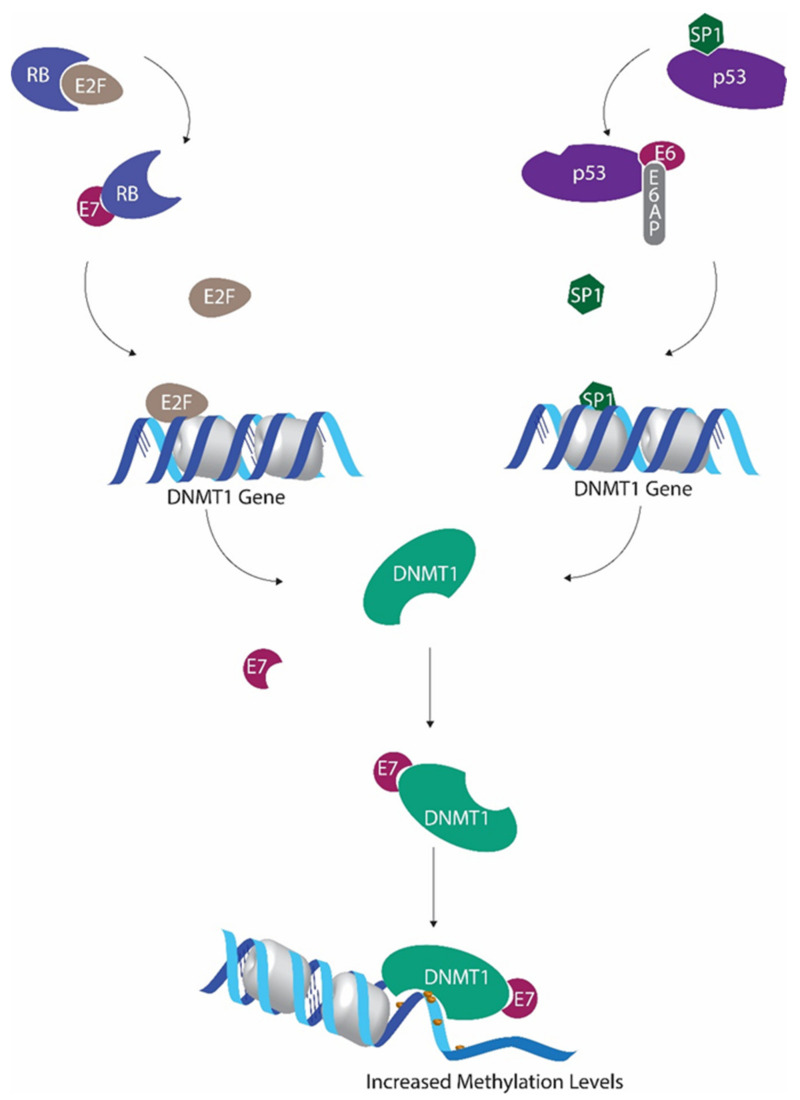
Human papillomavirus (HPV) E6 and E7 mediated regulation of methyltransferase activity. P53 inactivation by E6 leads to decreased Sp1 transcriptional factor inactivation, allowing increased DNMT1 transcription. E7-mediated downregulation of retinoblastoma protein hinders E2F sequestration. This E2F upsurge further promotes DNMT1 expression. E7 binds with DNMT1 to form the E7/DNMT1 complex, promoting DNMT1 activity.

**Figure 2 cells-09-01359-f002:**
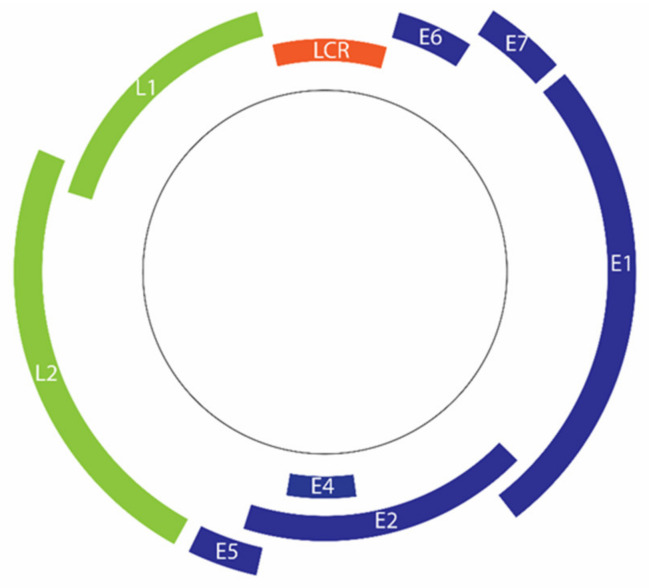
Organization of the HPV 16 genome. Deferent regions code for early genes (blue) and late genes (green). Transcription of early genes is regulated by the long control region (LCR) (Orange).

**Figure 3 cells-09-01359-f003:**
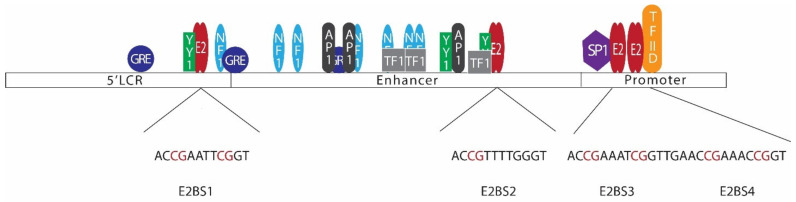
Arrangement of E2 binding sites in the HPV 16 long control region (LCR). The HPV 16 LCR can be subdivided into 3 sections, namely the 5’LCR, enhancer, and promoter regions. Binding of transcription factors such as YY1, NF1, GRE, AP1, SP1, TF1, and TFIID regulates viral early gene transcription. LCR consists of 4 E2 binding sites with a common palindromic sequence of ACCG(N)_4_CGGT (E2BS2 has a slight difference in the sequence). Several CpG sites are located in LCR E2 binding sites [138,140,141,142,143].

**Table 1 cells-09-01359-t001:** Differentially methylated genes and loci in HPV-driven head and neck cancer (HNC)**.**

Study	Samples	Type of Methylation Assessed	Methylation Status in HPV-Driven HNC		Comparison
			Hypermethylation	Hypomethylation	
Giuliano et al., 2019 [54]	Oropharyngeal cancer (OPC) (tissue and oral gargles)	*EPB41L3* methylation along with HPV DNA methylation	*EPB41L3* (both tissue and oral gargles)		Compared to control samples
	HPV-negative and positive controls (gargles)				
Ren et al., 2018 [55]	HPV-driven OPC (tissue)	Genome-wide DNA methylation	*KCNA3, EMBP1, CCDC181, DPP4, ITGA4, BEND4, CTNND2, ELMO1, SFMBT2, C1QL3, MIR129–2, ATP5EP2, OR6S1, NID2, HOXB4, ZNF439, ZNF93, VSTM2B, ZNF137P, ZNF773*		Compared to control samples
	Normal mucosal (tissue)				
Nakagawa et al., 2017 [46]	OPC (tissue)	Genome-wide DNA methylation	*GHSR, ITGA4, RXRG, UTF1, CDH8, FAN19A4, CTNNA2, NEFH, CASR*		Compared to HPV-negative OPC
	Normal mucosal (tissue)				
Swangphon et al., 2017 [56]	HNC (tissue)	*P16^INK4a^* promoter methylation	*P16^INK4a^* (90.9%)		Compared to control samples and HPV-negative HNC
	Normal oral (tissue)				
Esposti et al., 2017 [57]	HNC (tissue)	Genome-wide methylation	*ELMO1, CDH8, CRMP1, PCDH10, MSX2, SYN2, PCDHB11, HTR1E, PITX2, CDH18, CTNND2*	*SYCP2, RPA2, SMC1B, NCAN, NRXN1, COL19A1*	Compared to HPV-negative HNC
Lim et al., 2016 [58]	HNC (Saliva)	DNA methylation levels of *RASSF1α, p16^INK4a^, TIMP3, PCQAP/MED15* in Saliva		*p16 ^INK4a^, PCQAP/MED15*	Compared to control samples
	Control (Saliva)				
Worsham et al., 2016 [59]	HNC (tissue)	Methylation levels of 11 previously reported genes	*CDH8, PCDHB11, ELMO1, MSX2, HTR1E*	*MEI1 C14orf162/CCDC177*	Compared to HPV-negative HNC
Choudhury et al., 2015 [60]	HNC (tissue)	Methylation levels of 10 previously reported genes	*DAPK, RASSF1, p16, MINT31*		Compared to HPV-negative HNC
	Normal (tissue)				
Schlecht et al., 2015 [61]	OPC (tissue)	Genome-wide DNA methylation	22 CpG loci, including *CDKN2A* (non-promoter CpG), *GALR1, PPP1R3D*		Compared to normal samples
	Adjacent normal mucosal (tissue)				
Chen et al., 2015 [62]	OPC (tissue)	*IGSF4, DAPK1,* and *ESR1* promoter methylation	*IGSF4*		Compared to HPV-negative OPC
Kempen et al., 2014 [52]	OPC (tissue)	Promoter methylation—24 tumor suppressor genes	*CADM1, TIMP3*	*CHFR*	Compared to HPV-negative OPC
	Normal oropharyngeal (tissue)				
Kostareli et al., 2013 [63]	OPC (tissue)	Genome-wide DNA methylation	*BDNF, EOMES, GATA4, GFRA1, GRIA4, HOXA13, IRX4, PAX6, PHOX2B, SOX1, TBX5*	*ALDH1A2, FKBP4, GDNF, OSR2, PAX9, PROM1, PROX1, TLX1, UNCX, WIF1*	Compared to HPV-negative OPC
Weiss et al., 2013 [64]	HNC (tissue)	*TCF21* promoter methylation	*TCF21*		Compared to normal samples
	Benign tonsillar (tissue)				
Colacino et al., 2013 [65]	HNC (tissue)	Genome-wide DNA methylation	*CCNA1, GRB7, CDH11, RUNX1T1, SYBL1, TUSC3, GRPR, MC2R, GABRA5, PRSS1, NTSR1, F2R*	*SPDEF, RASSF1, STAT5A, MGMT, ESR2, JAK3, HSD17B12, CDK10, CHFR, RUNX3, APC, CDKN2A, STAT5A, JAK3, OSM, MPL, EPO*	Compared to HPV-negative HNC
Lechner et al., 2013 [53]	OPC (tissue)	Genome-wide promoter methylation	*CDH8, CDH15, PCDH8, PCDH9, PCDH10, PCDHB3, CDH13, CDH18, CDH19, CDH23, PCDH10, PCDH15, PCDHB1, PCDHB4, PCDHB15, PCDHB11*	*SNTB1, CYP7B1, MEI1, ICA1, FAM163A*	Compared to HPV-negative OPC
	Methylation data from other cancer types and cell lines				
Gubanova et al., 2012 [66]	HNC cell lines	*SMG-1* promoter methylation	*SMG-1*		Compared to HPV-negative OPC
	OPC (tissue)				
Sartor et al., 2011 [11]	HNC cell lines	Genome-wide DNA methylation	*LINE1, STS, ATP6V0C, HPS1, CTSL1, GNS, FUCA1, VPS18, IRS1, GNA11, GNAI2, EREG, CCNA1, RGS4, PKIG, KAL1, NF1, NGFR, GNAO1, SEMA3B, DCDC2, COL12a1, COL9a1, CYP2j2, GNAS, KAL1, MEST, RASGRF1, S1pr5, PREX1, RUNX2, SPON2, ESR1, DCC*	*PKCtheta, ESE3, RHOD, SOCS2, AnnexinIII, Annexin IX, TSPAN1, CDKN2A, CDKN2B*	Compared to HPV-negative HNC
	HNC (tissue)				
Weiss et al., 2011 [67]	HNC (tissue)	Promoter methylation of 12 genes	*CCNA1, TIMP3*		Compared to HPV-negative HNC
	Normal tonsillar (tissue)				
Poage et al., 2011 [68]	HNC (tissue)	Genome-wide DNA methylation	*LINE-1*		Compared to HPV-negative HNC
	Normal (tissue)				
Richards et al., 2009 [69]	Cancer cell lines	Methylation in *LINE* and *SINE (Alu)*	-		
	HNC (tissue)				
	Adjacent normal (tissue)				
Taioli et al., 2009 [70]	HNC (tissue)	Promoter methylation of *MGMT, CDKN2A*, and *RASSF1*		*CDKN2A*	Compared to HPV-negative HNC
Marsit et al., 2008 [71]	HNC (tissue)	*CDH1* promoter methylation	-		
O’Regan et al., 2008 [72]	HNC (tissue)	*CDKN2A (p16)* promoter methylation	-		
Furniss et al., 2008 [73]	HNC (tissue)	*LRE1* methylation	-		
Marsit et al., 2006 [74]	HNC (tissue)	Methylation of *SFRP1, SFRP2, SFRP4*, and *SFRP5*	*SFRP4*		Compared to HPV-negative HNC
Dong et al., 2003 [75]	HNC (tissue)	*RASSF1A* promoter methylation		*RASSF1A*	Compared to HPV-negative HNC
	HNC cell lines				

**Table 2 cells-09-01359-t002:** HPV DNA methylation in HPV-driven HNC.

Study	Samples	Type of Methylation Assessed	Methylation Patterns Reported
Giuliano et al., 2019 [54]	HPV-driven OPC (tissue and oral gargles)	CpG methylation in *L1, L2,* and *E2* genes	- Higher levels of methylation compared to HPV-positive controls - (*n* = 3)
	HPV-negative controls (gargles)		
	HPV-positive controls (gargles)		
Zhang et al., 2015 [147]	HNC and cervical cancer cell lines	CpG methylation in LCR	- LCR methylation; UM-SCC47: 79.8%; CaSki: 90%, SiHa; 0% - Lower levels of LCR methylation in OPC (overall 9.5%)
	HPV-driven OPC (tissue)		
Reuschenbach et al., 2015 [131]	HNC cell lines	Methylation of 10 CpGs in LCR	- > 80% methylation levels at E2BS3 and E2BS4 when HPV genome integration was present with intact *E2* genes. - 20%–80% methylation levels at E2BS3 and E2BS4 when episomal DNA is predominant. - < 20% methylation levels at E2BS3 and E2BS4 when *E2* gene is disrupted
	HPV-driven OPC (tissue)		
Wilson et al., 2013 [155]	HPV-driven HNC and non-HPV-driven HNC (tissue)	CpG methylation in the HPV genome	- Higher methylation level at the *L1–L2* boundary and within *E1*
	HNC cell lines		
Park et al., 2011 [148]	Cervical cancer cell lines	CpG methylation in the HPV genome	- CpG methylation—Cell lines CaSki: overall: 94%; LCR: 100% SiHa: overall: 35%; LCR: 0% - CpG methylation—OPC Low levels of methylation in LCR Similar methylation patterns in serum and saliva
	HPV-driven OPC (tissue, serum, saliva)		
Balderas-Loaeza et al., 2007 [149]	HPV-driven OC (tissue)	Methylation in 19 CpGs in *L1* gene and LCR	- Varying degrees of methylation in *L1* region and enhancer region - Low levels of methylation in LCR

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
