# Peer review of "DNA Methylation Changes in Human Papillomavirus-Driven Head and Neck Cancers"

_cells, 2020, doi:10.3390/cells9061359_

Round 1

Reviewer 1 Report

This manuscript deals with the DNA methylation changes in HPV-driven head and neck cancer. The authors give a brief but sufficient overview about the general features of HPV+ HNC and DNA methylation and then proceed to describing specific results regarding the methylation patterns of certain genes in HPV+ HNC and the viral genome itself. The paper is well written and provides an up to date overview. Results obtained so far are currently rather few, partly conflicting, suffering from different methodologies and often of unclear relevance but I have actually little to criticize regarding the review itself.

Only a few genes are described in more detail because they were identified in more than 1 study. As data are sparse, I was wondering whether there are maybe confirming or conflicting data regarding interesting candidates from studies on the methylation patterns in HPV-driven anogenital cancers as these have a similar biology? And another suggestion would be to briefly introduce 1 or 2 examples of tumor entities in which the methylation pattern is of relevance to underscore the principal potential of the topic.

Author Response

Comment

As data are sparse, I was wondering whether there are maybe confirming or conflicting data regarding interesting candidates from studies on the methylation patterns in HPV-driven anogenital cancers as these have a similar biology? And another suggestion would be to briefly introduce 1 or 2 examples of tumour entities in which the methylation pattern is of relevance to underscore the principal potential of the topic.

Response

As seen in HPV driven HNC, methylation patterns seem to be heterogeneous in other HPV driven cancers. As such indicating a few studies could be misleading. A comparison of these cancers would yield a lot of useful information as you have suggested. However, conducting an extensive comparison is a bit tedious process as there are over a hundred publications related to this topic in cervical, penile and anal cancers. Please note that we only considered studies investigating methylation patterns in the genes/loci which have been commonly reported in HPV driven HNC. We have included several examples form related cancers. Please note the modifications in line number; 189-190, 191-192, 194-195, 200-202, 223-224, 227-228, 234-235, 237-239, 261-262.

Reviewer 2 Report

In their review article entitled "DNA methylation changes in Human Papillomavirus driven Head and Neck Cancers" Ekanayake Weeramange and colleagues give an extensive overview of DNA methylation changes observed in HNC.

The review is well written and original, as the subject has not been extensively discussed by other publication recently. The article looks relevant for the HPV community since a clear view of the methylation landscape in HPV-associated cancer is far to be clear, as it correctly emerge from the present review. Indeed, as correctly highlighted by the authors, literature data are scattered and often controversial. This raise a question:  are there evidence that different HPV genotypes (e.g. HPV-16,18 or 33), could differently impact the methylation patter on HNC ? The author could include some data and/or make a comment about this point.

I was also wondering if preclinical data exist concerning demethylation approaches to treat HPV-associated HNC combined with standard treatment as radiotherapy (i.e. does demethylation treatment improve the response to RT or radio/chemotherapy?). This could be an important point to consider when transferring these approaches to the clinic.

Minor points:

Page 1 line 42: point is lacking at the end of the sentence

Page 4 line 139 please rewrite the sentence “E7 associated E2F increase further promotes DNMT1 expression.” Did the authors mean “E7-associated pRB downregulation frees E2F and further promotes DNMT1 expression” ?

Page 4 line 146 Please change expression to “activity”, as E2F is E7-expressing cells is released from its complex pRb

Page 10 line 275 should read double STRANDED DNA  instead of standard

Author Response

Comment

Are there evidence that different HPV genotypes (e.g. HPV-16, 18 or 33), could differently impact the methylation pattern on HNC?

Response

We agree with the comment. However, currently, there is not enough information regarding methylation changes caused by different HPV types. The majority of the studies have only considered HPV 16 as it is the most predominant type associated with HPV driven HNC. Those who have considered other HPV types have not described the changes in methylation patterns considering different HPV types. We have included a statement explaining the lack of data as suggested. Please refer to line 364-369.

Comment

I was also wondering if preclinical data exist concerning demethylation approaches to treat HPV-associated HNC combined with standard treatment as radiotherapy (i.e. does demethylation treatment improve the response to RT or radio/chemotherapy?).

Response

We could not find any completed clinical trials on the effectiveness of demethylation therapy for HPV driven HNC. A specific clinical trial to assess the effectiveness of demethylation therapy on HPV driven HNC is currently underway (NCT02178072) which is expected to be completed in May 2021. 

Comment

Page 1 line 42: point is lacking at the end of the sentence

Response

Corrected. (42)

Comment

Page 4 line 139 please rewrite the sentence “E7 associated E2F increase further promotes DNMT1 expression.” Did the authors mean “E7-associated pRB downregulation frees E2F and further promotes DNMT1 expression” ?

Response

Changed the sentence according to the suggestion (139-140)

Comment

Page 4 line 146 Please change expression to “activity”, as E2F is E7-expressing cells is released from its complex pRb

Response

Corrected. (147)

Comment

Page 10 line 275 should read double STRANDED DNA instead of standard

Response

Corrected (387)